

# The Hippo pathway in bone and cartilage: implications for development and disease

Chenwei Shao[1], Hao Chen[1,2], Tingting Liu[2] and Chun Pan[1,2]

[1] Institute of Translational Medicine, Yangzhou University, Yangzhou, Jiangsu, China
[2] Department of Orthopedics, Affiliated Hospital of Yangzhou University, Yangzhou, Jiangsu, China

## ABSTRACT

Bone is the main structure of the human body; it mainly plays a supporting role and participates in metabolic processes. The Hippo signaling pathway is composed of a series of protein kinases, including the mammalian STE20-like kinase MST1/2 and the large tumor suppressor LATS1/2, which are widely involved in pathophysiological processes, including cell proliferation, differentiation, apoptosis and death, especially those related to biomechanical transduction *in vivo*. However, the role of it in regulating skeletal system development and the evolution of bone-related diseases remains poorly understood. The pathway can intervene in and regulate the physiological activities of bone-related cells such as osteoclasts and chondrocytes through its own or other bone-related signaling pathways, such as the Wnt pathway, the Notch pathway, and receptor activator of nuclear factor-κB ligand (RANKL), thereby affecting the occurrence and development of bone diseases. This article discusses the role of the Hippo signaling pathway in bone development and disease to provide new insights into the treatment of bone-related diseases by targeting the Hippo signaling pathway.

## INTRODUCTION

The Hippo signaling pathway was originally found in Drosophila (*Harvey, Pfleger & Hariharan, 2003*) and has attracted great attention in recent years because of its high conservation in organisms and its ability to serve as a mechanosensitive factor. To date, many new Hippo pathway regulators have been discovered and identified (*Zheng & Pan, 2019*). Most recent studies have focused on its function in mammalian cells (*Fu et al., 2022*). Bone is an indispensable factor in the skeletal system that supports the normal physiological activities of organisms. Bone formation can be divided into two stages: the developmental stage (*Gkiatas et al., 2015*) and the remodeling stage (*Katsimbri, 2017*). The development of the skeleton can be traced back to the fetal period. The process is divided into intramembranous osteogenesis, endochondral osteogenesis, and, finally, mature bone (*Paiva & Granjeiro, 2014*). Destruction of the skeletal system can lead to different clinical diseases. These diseases include osteoporosis, which is caused by disorders of bone metabolism (*Yong & Logan, 2021*); arthritis due to destruction of chondrocytes (*Bessis et al., 2017*; *Konttinen et al., 2012*); and rheumatic diseases, which are caused by

Corresponding authors
Tingting Liu, apptx4869@126.com
Chun Pan, panchun0211@163.com

bone–immune imbalance (*McGonagle, 2010*). Therefore, a thorough understanding of the specific cellular signaling mechanisms and pathophysiological activities in various bone microenvironments can effectively prevent and treat bone-related diseases in clinical practice. The Hippo signaling pathway has been shown to be closely linked to bone-related diseases (*Wang et al., 2023*). Therefore, we focused on the Hippo signaling pathway to further investigate its relationship with bone diseases and discuss its potential in the treatment of bone-related diseases.

## SURVEY METHODOLOGY

### Objective

The objective of this survey methodology is to provide a comprehensive and unbiased overview of the literature pertaining to the role of the Hippo signaling pathway in bone. This includes understanding the pathway's involvement in various bone-related cells, its interaction with other signaling pathways, and its implications in the treatment of bone-related diseases.

### Search engines and databases

To ensure a thorough and exhaustive search, the following databases and search engines were utilized:

PubMed
Web of Science
Scopus
Google Scholar
Cochrane Library

These databases were chosen for their extensive coverage of scientific literature, including peer-reviewed articles, reviews, and conference proceedings.

### Search terms

A combination of keywords and Boolean operators was used to refine the search and ensure that all relevant literature was identified. The search terms included:

"Hippo signaling pathway"
"Bone development"
"Osteoblasts"
"Osteoclasts"
"Chondrocytes"
"Bone diseases"
"Wnt pathway"
"Notch pathway"
"RANKL"
"Osteoporosis"
"Arthritis"
"Rheumatic diseases"
"Osteosarcoma"

These terms were combined using "AND" and "OR" to create a comprehensive search strategy that captured the breadth of the topic.

## Inclusion and exclusion criteria

To maintain rigor and relevance, the following criteria were applied for the inclusion and exclusion of articles:

Inclusion criteria:

1. Articles published in English.

2. Peer-reviewed articles, reviews, and meta-analyses.

3. Articles that directly discuss the Hippo signaling pathway in the context of bone development or bone-related diseases.

4. Articles published within the last 15 years to ensure the currency of the information.

Exclusion criteria:

1. Non-peer-reviewed articles, editorials, opinion pieces, and letters to the editor.

2. Articles not directly related to the Hippo signaling pathway or bone-related diseases.

3. Articles published in languages other than English.

4. Duplicate publications or studies.

## Search strategy

The search strategy involved an initial broad search using the primary keywords, followed by a more targeted search using the refined terms. The titles and abstracts of the identified articles were screened to assess their relevance. Full texts of potentially eligible articles were obtained and assessed against the inclusion and exclusion criteria.

## Data extraction and analysis

Data extraction was performed by two independent reviewers to minimize bias. Discrepancies were resolved through discussion or by involving a third reviewer. The extracted data included author, year of publication, study design, main findings, and conclusions. The data were then synthesized to provide a comprehensive overview of the current state of research on the Hippo signaling pathway in bone.

## Quality assessment

To ensure the quality of the included studies, a critical appraisal was conducted using standardized tools such as the Cochrane Risk of Bias Tool for randomized controlled trials and the Newcastle-Ottawa Scale for observational studies. This assessment helped to evaluate the methodological rigor and reliability of the included studies.

## Transparency and reproducibility

To maintain transparency and reproducibility, a detailed record of the search history, including the search terms, databases searched, and the number of articles identified, included, and excluded at each stage, was kept. This record will be made available.

By following this rigorous and systematic approach, we aim to provide an unbiased and comprehensive coverage of the literature on the Hippo signaling pathway in bone.

## THE INTENDED AUDIENCE

The document "Hippo Pathway in Bone" is intended for a specialized audience with a strong background in cellular and molecular biology, particularly those with interests in the skeletal system, bone metabolism, and related diseases. The specific audience can be categorized as follows:

1. Researchers and scientists: Professionals engaged in the study of bone biology, including the development, growth, and remodeling of bones, as well as the molecular mechanisms underlying bone-related diseases.

2. Clinicians and medical professionals: Orthopedic surgeons, rheumatologists, and other clinicians who diagnose and treat bone and joint diseases, such as osteoporosis, arthritis, and osteosarcoma.

3. Pharmacologists: Those involved in the development of drugs targeting the Hippo signaling pathway for the treatment of bone diseases, including osteoporosis and potentially osteosarcoma.

4. Graduate students and academics: Students pursuing advanced degrees in biology, medicine, or related fields, as well as academics teaching courses on cellular signaling, bone biology, and related topics.

5. Biotechnology and pharmaceutical industry professionals: Individuals in the biotech and pharmaceutical sectors who are interested in the therapeutic potential of targeting the Hippo signaling pathway for bone diseases.

6. Regulatory agencies and policy makers: Professionals in health authorities and regulatory bodies who may influence or make decisions regarding the approval and regulation of new treatments for bone diseases.

7. Patient advocacy groups: Organizations and individuals advocating for patients with bone diseases, who may use this information to better understand their conditions and potential new treatments.

The document's technical nature and focus on the molecular details of the Hippo signaling pathway in bone suggest that it is not intended for the general public but rather for those with a professional or academic interest in the field.

### Hippo signaling pathway

The Hippo signaling pathway is composed of a series of different protein kinases (*Ma, Meng & Guan, 2019*). In mammals, it is mainly composed of three core parts, namely mammalian STE20-like kinase 1/2 (MST1/2) and its ligand Salvador homologue 1 (SAV1), This was followed by large tumor suppressor kinase 1/2 (LATS1/2) and its regulatory protein MOBKL1A/B (MOB1A/B) in the middle part. The downstream effector molecules Yes-associated protein 1 (YAP) and WW-domain-containing transcription regulator 1 (TAZ) are also involved (*Wang & Martin, 2017*). When the Hippo pathway is activated, MST1/2 and LATS1/2 are phosphorylated by their regulatory proteins Sav and MOB, which phosphorylate the downstream effectors YAP/TAZ, which are retained in the cytoplasm and either bind to 14-3-3 or are ubiquitylated. It further prevents its translocation into the nucleus (*Driskill & Pan, 2021*). However, when the Hippo signaling pathway is turned off, YAP/TAZ phosphorylation is inhibited, and YAP/TAZ is transferred to the nucleus, where

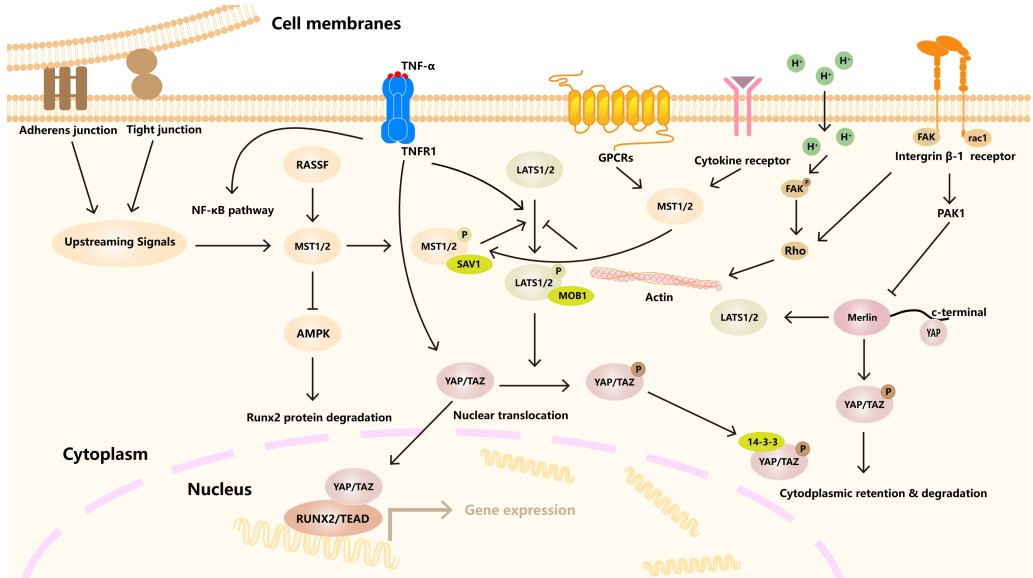

**Figure 1** **Hippo signaling pathway regulates cell homeostasis under many factors.** A variety of receptors, channels, and signaling molecules, such as GPCRs, integrin, TNFR1. The activation of LATS1/2 kinase by MST1/2 kinase, which in turn phosphorylates YAP/TAZ transcriptional coactivator. Downstream effects include YAP/TAZ binding to TEAD transcription factors to regulate gene expression. In addition, other signaling pathways and cytoskeleton, connectivity and other factors are involved.

it accumulates and interacts with the TEA domain (TEAD), runt-related transcription factor (RUNX), Smad and other binding proteins and then binds to DNA to regulate the transcription of related target genes, thereby producing biological effects (*Wang & Martin, 2017*). Many regulatory pathways are upstream of the Hippo signaling pathway, such as MAP4Ks (mitogen-activated protein kinase kinase kinase kinase), RASSFs (Ras association domain family member), and NF2 (Neurofibromatosis type II) (*Wang & Martin, 2017*; *Yin et al., 2013*) (Fig. 1). RASSF is an upstream key regulatory molecule of the Hippo signaling pathway that can bind to MST to form RASSF/MST homodimer to induce the homodimerization, phosphorylation and activation of MST1/2. These series of activities play an important role in regulating the biological functions of cells (*Oceandy et al., 2019*; *Bitra et al., 2017*).

## Hippo in bone development

Bone development is a complex process, starting from the fetal period. Initiation of bone formation begins with the migration of mesenchymal cells derived from a range of embryonic lineages to future skeletal sites (*Hall & Miyake, 1992*). Then, it undergoes a series of intramembranous and endochondral osteogenesis processes. Finally, a mature skeleton is formed. As a key signaling pathway regulating the growth and development of organisms (*Berendsen & Olsen, 2015*), the Hippo signaling pathway plays an indispensable role in the formation and growth of bones (Table 1).
**Table 1  Key signaling molecules in various bone cells.**

| Component | Cell type | Pathologic role | Reference |
|---|---|---|---|
| MST1/2 | Osteoblasts | MST2 upregulates RUNX2, ALP, promoting osteoblast precursor maturation. MST1/2 knockout mice show decreased RUNX2 expression. | *Lee et al. (2015)*, *Larsson et al. (2008)* |
| | BMSCs | MST1 is a key factor in bisphosphonate-mediated inhibition of bone resorption. MST2 deficiency results in increased RANKL-induced osteoclast precursor differentiation. | *Lee et al. (2015)*, *Song et al. (2012)* |
| LATS1/2 | Osteoblasts | LATS1/2 phosphorylates YAP/TAZ, regulating osteoblast differentiation and function. | *Wang & Martin (2017)* |
| YAP/TAZ | Osteoblasts | YAP/TAZ knockdown decreases osteoblast activity; YAP promotes osteoblast progenitor proliferation and differentiation, inhibits adipogenesis, and positively regulates β-catenin signaling. | *Li et al. (2024a)*, *Yang et al. (2020)* |
| | Chondrocytes | YAP dephosphorylation and nuclear localization inhibit chondrocyte development and maturation via different pathways, including suppression of Sox9. | *Deng et al. (2016)*, *Goto et al. (2018)* |
| | Osteoclasts | YAP1 is essential for osteoclastogenesis through a TEADs-dependent mechanism. YAP/TAZ knockdown inhibits osteoclast formation. | *Li et al. (2024a)*, *Yang et al. (2021a)* |
| | BMSCs | YAP expression in BMSCs is associated with migration and osteogenesis-related abilities. YAP/TAZ bind to SNAIL/SLUG, regulating MSCs and osteogenic differentiation. | *Wang et al. (2019)*, *Jensen et al. (2015)*, *Reszka et al. (1999)* |
| SAV1 | Osteoblasts | SAV1 indirectly affects osteoblast differentiation via MST1/2 and LATS1/2. | *Wang & Martin (2017)* |
| MOB1A/B | Osteoblasts | MOB1A/B regulates LATS1/2 activity, influencing YAP/TAZ phosphorylation and osteoblast function. | *Wang & Martin (2017)* |
| NF-κB | Chondrocytes | YAP attenuates OA progression by inhibiting NF-κB signaling-mediated inflammatory responses. | *Gong et al. (2019)* |
| RUNX2 | Osteoblasts | RUNX2 expression is upregulated by MST2 and YAP/TAZ, promoting osteoblast differentiation. RUNX2 is decreased in MST1/2 knockout mice. | *Lee et al. (2015)*, *Larsson et al. (2008)*, *Yang et al. (2020)* |
| RANKL | Osteoclasts | TAZ inhibits RANKL-induced osteoclast maturation; TAZ gene expression is lower in osteoporotic mice. RANKL activation reduces TAZ, which can be reversed by MG132. | *Xiong, Almeida & O'Brien (2018)* |
| Smad4 | Osteoblasts | TAZ binds with Smad4 to co-stimulate RUNX2 in precursor osteoblasts, promoting MSC differentiation. | *Yang et al. (2020)* |
| VGLL3 | RA-FLSs | VGLL3 knockdown increases TAZ expression, while VGLL3 overexpression suppresses TAZ. VGLL3 activates the Hippo pathway, promoting cell proliferation. | *Wang et al. (2020)*, *Tomás et al. (2016)* |
| SNAIL/SLUG | BMSCs | SNAIL/SLUG bind to YAP/TAZ, regulating MSC self-renewal and differentiation; SNAIL/SLUG knockout inhibits YAP/TAZ-TEAD-related gene expression. | *Reszka et al. (1999)*, *Li et al. (2018)* |

## Hippo in different bone cells

### Hippo in bone stem cells

As the origin of osteoblasts, stem cells play an indispensable role in bone formation. The Hippo signaling pathway is involved in the development of bone stem cells. Most Prx1-positive mesenchymal stem cells (MSCs) contain the Hippo signaling pathway (*Gou et al., 2024*). Piezo1 is a nonselective calcium channel located in osteoclasts that senses mechanical load and promotes the translocation of YAP into the nucleus. The loss of Piezo1 or Piezo1/2 in MSCs or osteoblast progenitor cells can inhibit the formation of the NFAT/YAP1/β-catenin complex, hinder the development and maturation of osteoblasts, and cause bone loss in mice (*Zhou et al., 2020*). TAZ expression in adipose-derived stem cells (ADSCs) increases osteogenic differentiation as well as bone regeneration *in vivo* (*Zhu et al., 2018*). When the osteogenesis-related gene GNAS was knocked out in mesenchymal stem cells (MSCs), the activation of the Hippo signaling pathway was promoted, and the differentiation of osteoblasts was inhibited (*An et al., 2019*). Moreover, Hippo can also regulate the renewal and differentiation of embryonic stem cells and induce pluripotent stem cells by affecting a variety of other transcription factors, thereby affecting the development of craniofacial bones (*Xiang et al., 2018*). Hippo can regulate the development and differentiation of MSCs, but how it affects the development and differentiation of bone marrow stem cells needs further study.

### Hippo in cartilage cells

The development and differentiation of chondrocytes are mediated by cell molecules, mechanical transduction, hormones and other signals (*Long & Ornitz, 2013*). TNC deposition in the endochondral environment, through mechanical signaling, leads to the inactivation of YAP and enhances chondrocyte differentiation (*Li et al., 2021b*). In addition, the dephosphorylation of YAP is accompanied by nuclear localization and osteogenesis-related gene RUNX2, which inhibits the expression of Col10a1, thereby inhibiting the development and maturation of chondrocytes (*Deng et al., 2016*). The nuclear localization of YAP inhibits Sox9 and arrests chondrocyte growth and development (*Goto et al., 2018*). The dephosphorylation and nuclear localization of YAP inhibit the occurrence and development of chondrocytes and hinder bone formation through different pathways.

### Hippo in osteoblasts

Osteoblasts originate from mesenchymal stem cells in the mesoderm of the embryonic stage. The inhibition of YAP decreases bmp2a expression in MSs, leading to the accumulation of osteoprogenitor cells (*Brandão et al., 2019*). The migration-related and osteogenesis-related abilities of bone marrow mesenchymal stem cells are associated with increased expression of YAP (*Wang et al., 2019*) (Fig. 2).

In addition, MST2, a key upstream signaling molecule of the Hippo signaling pathway, is involved in the regulation of osteogenesis. *Lee et al. (2015)* reported that ectopic expression of MST2 in osteoblast precursors upregulated the expression of osteoblast-related genes such as RUNX2 and ALP, thereby increasing the transformation of osteoblast precursors into mature osteoblasts.

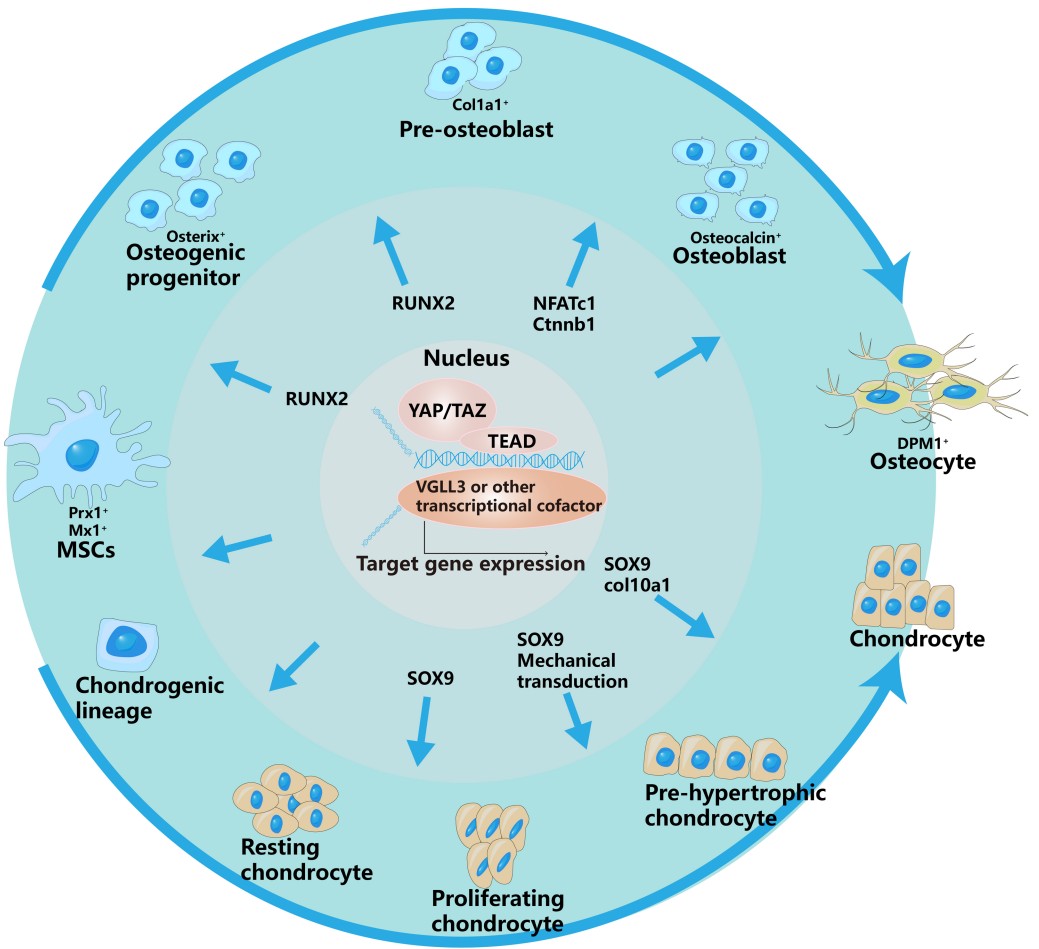

**Figure 2** **YAP/TAZ have different regulatory effects on different stages of osteogenic and chondrogenic differentiation.** This is specifically controlled by interacting with different signaling pathways and regulating different transcription factors.The outer arrows in the picture represent the different differentiation directions of osteoblasts and chondrocytes, and the inner arrows represent the regulation of YAP/-TAZ on different differentiation stages.

### Hippo in osteoclasts

Hippo signaling pathway plays an important role in the occurrence, differentiation and function regulation of osteoclasts (Fig. 3). The core mechanism involves the activity regulation of downstream effector molecules YAP/TAZ and its interaction with other signaling pathways (*Yang et al., 2018*). RASSF2 interacts with MST1/2 and stabilizes its activity. *Song et al. (2012)* found that BMMs from RASSF2$^{-/-}$ mice form more TRAP positive multinucleated cells, and can enhance the bone resorption ability of osteoclasts. The NF-κB signaling pathway plays a key role in osteoclast differentiation and function. In RASSF2$^{-/-}$ mice, the NF-κB signaling pathway was over-activated in osteoclast precursor cells, as indicated by the significantly increased phosphorylation level of IkB-α. Osteoclasts are derived from hematopoietic stem cells (HSCs). NF2-deficient mice have a significant increase in the number of HSCs and promote the translocation of HSCs to the bone

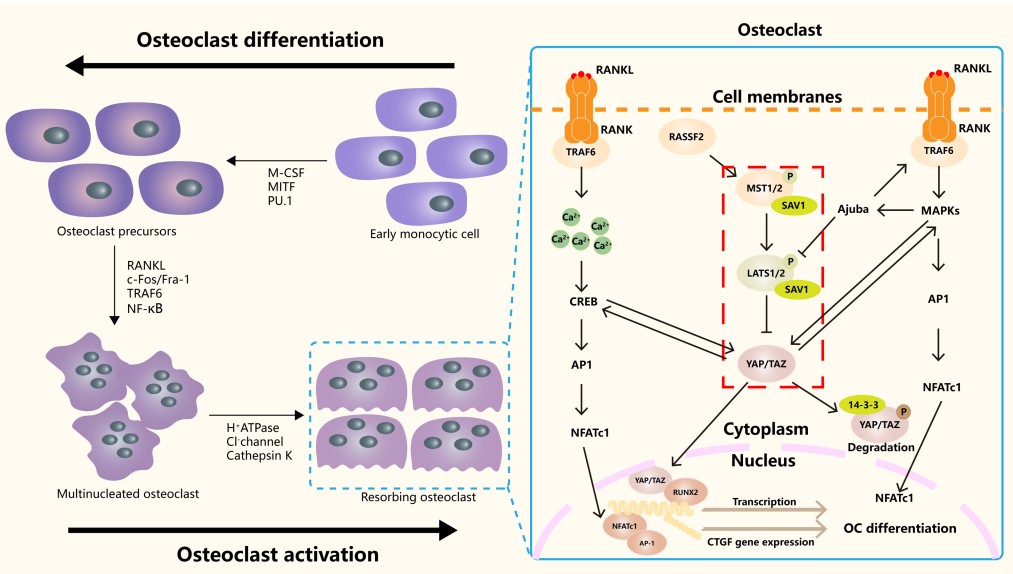

**Figure 3** **The role of the Hippo signaling pathway in osteoclasts.** The Hippo signaling pathway as a key regulatory axis influencing osteoclast differentiation and activity. YAP/TAZ interact with NFATc1. This synergy amplifies NFATc1-mediated gene expression, enhancing osteoclast differentiation. The RANKL-TRAF6 axis (*via* MAPKs/NF-κB) may modulate Hippo activity, suggesting bidirectional regulation. Nuclear YAP/TAZ promote expression of genes like CTGF, which facilitates cytoskeletal organization and bone-resorbing capacity in mature osteoclasts.

marrow. Although the relationship between NF2 and Hippo signaling remained directly explored, these changes may regulate the behavior of HSCs by affecting some components of the Hippo signaling pathway. It also affects the growth and development of osteoclasts (*Larsson et al., 2008*). It was found that cartilage endplate cells release CCL3, a key factor in osteoclast recruitment and activation, in response to abnormal stress. The expression of CCL3 is regulated by YAP, and the activation of YAP can inhibit CCL3 expression, thereby inhibiting osteoclast maturation (*Li et al., 2024a*). In addition, YAP/TAZ has been found to inhibit the activation of NF-κB signaling pathway by binding to TAK1, thereby hindering osteoclast differentiation (*Yang et al., 2020*). The RANKL-RANK-OPG system is a major mechanism regulating osteoclast differentiation, activation, and survival. YAP/TAZ also affected OC differentiation by regulating OPG expression. Upregulation of YAP/TAZ promotes OPG expression, which acts as a decoy receptor for RANKL and inhibits RANKL-RANK interaction, thereby reducing OC differentiation and bone resorption (*Yang et al., 2021a*).

### Hippo in osteoporosis

Previous studies have demonstrated that the Hippo signaling pathway is closely related to bone metabolism and that the stability of bone metabolism is based on the balance of the osteogenic and osteoclast microenvironments. On the one hand, the Hippo signaling pathway directly mediates bone metabolism by regulating bone-related cells. On the other hand, by interacting with other signaling pathways, such as the NOTCH, Wnt/β-catenin,

and NF-κB pathways, it indirectly regulates the growth and development of osteoblast-osteoclast and controls the balance of the osteoblast-osteoclast microenvironment. Disorders of bone metabolism can lead to a series of metabolic bone diseases, such as osteoporosis (*Jensen et al., 2015*).

## Regulation of MST on bone remodeling

As an upstream protein kinase of the Hippo signaling pathway, MST plays an indispensable role in cell proliferation and differentiation, but its specific mechanism has not been fully studied. MST1/2 is an upstream molecule of the Hippo signaling pathway. Previous studies have confirmed that MST1 is a key factor in the inhibition of bone resorption by bisphosphonates (*Reszka et al., 1999*). However, whether MST1/2 controls osteoclast development by regulating YAP1 activity needs to be further confirmed. Notably, MST2 deficiency results in increased RANKL-induced osteoclast precursor differentiation, and MST2 KO mice exhibit an osteoporotic phenotype with increased osteoclasts and decreased osteoblasts (*Lee et al., 2015*). The possible mechanism is that MST2 deficiency downregulates the expression of the osteogenesis-related genes RUNX2, ALPL, and LBSP. MST1/2 exerts a multifaceted role in regulating bone development and remodeling. The expression of RUNX2, an osteogenesis-related transcription factor, was also decreased in MST1/2 knockout mice (*Li et al., 2018*). A possible explanation is that MST1/2 may control the occurrence and outcome of osteoporosis by regulating bone metabolism.

## YAP/TAZ regulation of bone remodeling

As key downstream signaling molecules of the Hippo signaling pathway, YAP/TAZ play important roles in bone remodeling. At the cellular level, YAP/TAZ knockdown decreases the activity of osteoblasts and increases the activity of osteoclasts and their relationship showed a dose-dependent manner (*Kegelman et al., 2018*). YAP has the potential to regulate and promote the proliferation and differentiation of osteoblast progenitor cells, inhibit the adipogenesis of mesenchymal stem cells, and play a positive regulatory role in the β-catenin signaling pathway during osteogenesis and bone homeostasis. In addition, TAZ binds with Smad4 to co-stimulate RUNX2 in precursor osteoblasts and promote the differentiation of MSCs into osteoblasts (*Park et al., 2019*). *Zhao et al. (2018)* first reported that, in osteoclasts, YAP1 is involved in the regulation of osteoclastogenesis. Knockdown of YAP1 strongly inhibited the formation of osteoclasts. In addition, YAP/TAZ can bind to SNAIL and SLUG to regulate MSCs and further affect osteogenic differentiation (*Guo et al., 2012*). *Tang et al. (2016)* reported that the reduction in SNAIL1/SNAIL2 could inhibit the expression of YAP/TAZ-TEAD-related genes. The terminal differentiation program was blocked in SNAIL/SLUG double-knockout MSCs. The increased phosphorylation of YAP/TAZ caused by a lack of SNAIL/SLUG is closely related to this (*Tang & Weiss, 2017*). YAP/TAZ is not only affected by upstream molecules of the classic Hippo pathway but also regulated by various upstream signals related to osteogenesis, playing a vital role as intermediate nodes.

Interestingly, *Xiong, Almeida & O'Brien (2018)* reported that YAP and TAZ inhibited the canonical Wnt/β-catenin signaling pathway and RUNX2 activity in osteogenic progenitor
cells. However, the levels of regulatory molecules related to bone turnover, such as RANKL and OPG, do not change. Consistent with these findings, the inhibition of YAP in MSCs partly promoted osteoblast differentiation *in vitro* (*Seo et al., 2013*). *In vivo*, the inhibition of TAZ expression in zebrafish hindered osteoblast differentiation (*Jeong-Ho Hong 1 et al., 2005*). These results suggest that YAP and TAZ in osteogenic progenitor cells exert inhibitory effects on their ability to differentiate into osteoblasts, but in mature osteoblasts and osteocytes, they instead promote bone formation and inhibit bone resorption. However, it is clear that YAP/TAZ binds to TEAD1 and TEAD4 in the nucleus to form a transcription complex, which activates AP1 and NFATc1 to regulate the expression of transcription factors in osteocytes. Moreover, YAP and TAZ interact with other transcription factors, such as RUNX2, CREB and AP1, which further enriches our understanding of the regulation of downstream genes in the Hippo pathway (*Wang et al., 2023*). These findings provide new insights into the temporal role of YAP/TAZ in regulating bone-associated cells.

YAP and TAZ are closely related to the occurrence and development of osteoporosis. A number of studies have reported that the Hippo signaling pathway is regulated by miRNAs and subsequently produces biological effects. In a study by *Li et al. (2021a)*, exosomes derived from human bone marrow mesenchymal stem cells promoted osteogenesis in ovariectomized rats by increasing MOB1 and YAP and transferring miR-186. In addition, myoblast-derived exosomal Prrx2 can also activate the YAP pathway through miR-128 to promote the osteogenic differentiation of MSCs (*Li et al., 2023b*). In addition, *Li et al. (2023)* reported that the inhibition of YAP expression inhibited osteoclast maturation and survival regardless of the overexpression of 11β-hydroxysteroid dehydrogenase type 1 (11β-HSD1), a substance that promotes osteoclast maturation, effectively protected bone and reduced the degree of osteoporosis in ovariectomized mice. The upregulation and downregulation of YAP expression are regulated by many proteins and thus play indispensable roles in the development of osteoporosis.

## Effects of YAP/TAZ interactions with other signaling pathways on osteoporosis

TAZ, the downstream effector of the Hippo signaling pathway, is closely related to the development and maturation of osteoclasts induced by RANKL. TAZ gene expression was significantly lower in osteoporotic mice than in normal control mice. TAZ inhibited RANKL-induced osteoclast maturation *in vitro*. These findings suggest that TAZ is negatively correlated with osteoclast development and may act with TAK1 to downregulate the NF-κB signaling pathway and further osteoclast differentiation (*Yang et al., 2021a*). The reduction in TAZ caused by RANKL activation can be reversed by MG132 (a 26S proteasome inhibitor), suggesting that TAZ, a downstream signaling molecule in the Hippo signaling pathway, may be a new target for the treatment of osteoporosis.

## Hippo in cartilage related diseases
### Hippo in arthritis
Osteoarthritis (OA) is a chronic joint disease characterized by degenerative changes in the articular cartilage and secondary hyperosteogeny (*Xia et al., 2014*; *Liu et al., 2022b*). The disease affects the articular cartilage or the entire joint, including the subchondral

bone (*Cardoneanu et al., 2022*), the joint capsule (*Sanchez-Lopez et al., 2022*), the synovium (*Liu et al., 2022a*), and the muscles surrounding the joint. It is of great physiological and pathological significance to understand the progress of Hippo in OA.

### Effect of Hippo on articular cartilage

The expression of YAP is elevated in articular cartilage or chondrocytes in both mouse and human OA (*Gong et al., 2019*). *Bottini et al. (2019)* demonstrated that YAP increases TGF-β-dependent SMAD3 nuclear localization and that the inhibition of YAP improves OA symptoms. *Zhang et al. (2022)* reported that RUNX1 may inhibit OA by controlling the amount of the YAP protein and coordinating many signaling pathways, including the Wnt, Hippo, and TGF pathways. T-2 Mycotoxin damages cartilage, leading to a disease phenotype similar to osteoarthritis, and *Li et al. (2022)* reported that T-2 Mycotoxin reduces COL2 and PCNA levels in chondrocytes after the inhibition of YAP. Previous studies have suggested that the NF-κB signaling pathway plays an indispensable role in the development and progression of arthritis. Because when joint inflammation occurs, the development of chondrocytes is regulated by many different types of cytokines. Cytokines such as tumor necrosis factor-α (TNF-α), interleukin-1β (IL-1β), and IL-6 play regulatory roles in the occurrence of arthritis (*Yasuda, 2011*). Each of these cytokines can further interfere with the activation or inhibition of the NF-κB signaling pathway (*Kapoor et al., 2011*), thereby promoting arthritis. *Deng et al. (2018)* reported that Hippo signaling controls articular cartilage homeostasis through YAP-mediated NF-κB signaling. YAP can attenuate the progression of OA by inhibiting the inflammatory response triggered by the NF-κB signaling pathway. In addition to participating in disease progression, differentiated chondrocytes also inhibit cartilage hypertrophy by increasing the expression of YAP1 (*Ying et al., 2018*).

### Effect of Hippo on subchondral bone

Subchondral bone, below the calcified cartilage, maintains and distributes the mechanical stress caused by exercise and loading (*Lories & Luyten, 2011*). Studies on the effect of Hippo on subchondral bone development are scarce. However, "dentine changes" in subchondral bone are critical in the pathological process of OA. Notably, *Xiao et al. (2023)* reported that increased miR-6215 expression mediates the dexamethasone (PDE)-induced decrease in subchondral bone mass *via* FRMD6/YAP1 signaling. However, more studies are needed to explore the role of Hippo in this process.

### Effect of Hippo on synovium

Synovial lesions are associated with pain and pathological responses caused by OA and are the typical disease characteristics of OA. In mouse models of antigen-induced arthritis (AIA), inhibition of YAP affects synovial vasculogenesis (*Chen et al., 2021*) and leads to rheumatoid arthritis (RA) progression. Leonurus regulates the Hippo signaling pathway through the miR-21/YOD1/YAP axis to reduce joint inflammation and bone destruction in CIA mice, thereby inhibiting the growth and inflammation of rheumatoid arthritis fibroblast-like synoviocytes (RA-FLSs) (*Ma et al., 2024*). In OA, *Deng et al. (2018)* reported increased numbers of inflammatory cells and synovial lining cells in the synovium of

Yap$^{fl/fl}$ mice. However, further evidence on how YAP modulates synovial cell progression in OA is lacking.

### Hippo in degenerative diseases of spine

Intervertebral disc degeneration is the main cause of low back pain. Despite the long history of intervertebral disc degeneration, its onset and progression are poorly understood, and a standard model of disease occurrence is lacking (*Murray et al., 2012*). Degenerative disc deformation and injury leading to changes in mechanical and biological behavior are important causes of pathological symptoms (*Vo et al., 2016*). The etiology of natural lumbar disc degeneration (IDD) is complex, with mechanical, inflammatory, and structural causes contributing to increased pain and worsening patients' symptoms (*Feng, Egan & Wang, 2016*).

The intervertebral disc (IVD) is composed of heterogeneous cell populations, including those of the nucleus pulposus (NP) and annulus fibrosus (AF), which undergo different ECM and mechanical loads. As a classical signaling pathway related to mechanical stress, the correlation between Hippo and mechanical stress deserves further exploration. Natural IDD patients show a time-dependent decrease in YAP activity with F-actin (*Zhang et al., 2021*). *Zhang et al. (2018)* reported that the expression of YAP in cartilage endplate cells (CEs) was consistent with that in NP cells and gradually decreased with age in disc-depleted IDD model mice, suggesting that YAP was positively correlated with IDD repair. In a rat IVDD model, the overexpression of SEPHS1 and the inhibition of Hippo–YAP/Taz attenuated the progression of IVDD (*Hu et al., 2024*). The LAT/YAP/CTGF signaling pathway promotes the synthesis and catabolism of the ECM in nucleus pulposus cells (NPCs), inhibits the catabolism of the ECM, and delays the progression of IDD (*Chen et al., 2022*). Matrix stiffness-induced ferroptosis in NP cells can be reversed by inhibiting the nuclear translocation of YAP (*Ke et al., 2023*). However, a specific inhibitor of YAP1, verteporfin, can effectively alleviate IDD development in rat intervertebral discs (*Chen et al., 2019*). *Croft et al. (2021)* reported that extranuclear YAP was observed only under weak loading conditions, that is, under static loading and low-stress conditions. The increased activation and subsequent nuclear localization of YAP under stronger loading conditions confirmed that mechanical stress can trigger the translocation of YAP and TAZ to promote gene expression (*Li et al., 2024b*).

### Hippo in rheumatism

Bone metabolism, growth, development, remodeling and repair are inseparable from the interaction between osteoblasts and osteoclasts. Immune-related active components such as T cells, B cells, dendritic cells, Toll-like receptors, chemokines, and interleukins cooperate with osteoclasts to regulate bone formation and resorption, thereby changing the direction of bone formation (*Wang et al., 2020*). Autoimmune diseases occur when the body loses immune homeostasis. Rheumatoid arthritis (RA) is a complex chronic inflammatory disease in which multiple joints in the body become inflamed. Synovitis continues to worsen over time (*Tomás et al., 2016*), with joint degeneration and deformity and loss of function (*Cross et al., 2014*).

*Chen et al. (2021)* used antigen-induced arthritis mice. Silencing Ezrin prevents synovitis in these animals by downregulating the phosphorylation of YAP and preventing its nuclear accumulation. Previous studies have demonstrated that YAP may exert some control over the PI3K/Akt pathway (*McLoughlin, Mueller & Grossmann, 2018*) and that activated YAP stimulates the PI3K/Akt pathway, promotes the proliferation and activity of vascular endothelial cells, and regulates the growth of synovial arteries in mouse RA and the inflammatory process in RA joints. Berberine, a traditional Chinese medicine, inhibits the pyroptosis of RA-FLSs by downregulating the NLRP3 inflammasome and alleviating collagen-induced arthritis by activating the Hippo signaling pathway (*Zhang et al., 2024*).

Degenerate family member 3 (VGLL3) is a homolog of degenerate-like genes in Drosophila. It may be a transcriptional cofactor of TEADs (*Mesrouze et al., 2020*). *Du et al. (2022)* reported that knockdown of VGLL3 increased TAZ expression but had no effect on YAP expression in RA-FLSs. On the other hand, TAZ expression was suppressed by VGLL3 overexpression. Furthermore, VGLL3 activates the Hippo pathway and promotes the proliferation of A549 cells and MDA-MB-231 cells (*Hori et al., 2020*).

### Hippo in osteosarcoma

Osteosarcoma is characterized by a poor prognosis, difficulty in treatment, and poor patient survival, whereas the most common primary osteosarcoma, for example, primarily affects children, adolescents, and young adults, with the second highest incidence among the elderly (*Corre et al., 2020*). The Hippo signaling pathway further controls the occurrence and development of osteosarcoma. In summary, the process can be divided into three main aspects: the dysregulation of upstream regulatory factors, the abnormal activation of YAP/TAZ, and the interaction with the tumor microenvironment (*Rothzerg et al., 2021*).

In osteosarcoma cells, the downregulation of MST1/2 and LATS1/2 leads to reduced phosphorylation of YAP/TAZ, thereby increasing the nuclear translocation and activity of YAP/TAZ (*Chan et al., 2011*). Sox2, as an upstream signaling molecule of the Hippo signaling pathway, inhibits the activation factors NF2 (neurofibromin 2) and KIBRA of the Hippo pathway, resulting in decreased YAP phosphorylation and subsequent nuclear accumulation, which promotes the development and progression of osteosarcoma (*Basu-Roy et al., 2015*). *Wu et al. (2020)* reported that in SRPX2-knockdown cells, decreased YAP phosphorylation and reduced YAP protein levels were detected in 143B and U2 osteosarcoma cells. In summary, the dysregulation of the upstream regulators leads to the accumulation of YAP/TAZ, which leads to the abnormal proliferation and differentiation of cells and promotes the occurrence and development of osteosarcoma.

In osteosarcoma, YAP/TAZ is frequently overexpressed and is associated with tumor aggressiveness and poor prognosis. The nuclear localization of YAP/TAZ is significantly increased, playing a crucial role in the progression of osteosarcoma through various mechanisms. MST1 and YAP promote the expression of PLOD2 and tumor cell migration in human osteosarcoma cell lines (*Trang et al., 2023*). YAP/TAZ interacts with the TEAD transcription factors to activate the expression of downstream genes, thereby promoting cell proliferation and inhibiting apoptosis (*Chai, Xu & Guo, 2017*). Studies (*Zanconato et al., 2015*) have indicated that mitogen-activated protein kinase (MAPK) is closely related

to YAP in the Hippo signaling pathway, and that AP1 is a downstream signaling molecule of the MAPK pathway. This interaction can enhance the formation of the YAP-TEAD complex, thereby promoting the expression of downstream YAP-TEAD complex-related genes, significantly increasing the growth and development of tumor cells and enhancing their tumorigenicity. Verteporfin attenuates the interaction between YAP1 and TEAD1 (*Yang et al., 2021b*). Additional studies have shown that verteporfin delays the progression of osteosarcoma in Ctsk-Cre; Trp53f/f/Rb1f/f mice by inhibiting YAP/TAZ and preventing bone erosion, providing new possibilities for the treatment of osteosarcoma (*Li, Yang & Yang, 2022*). Moreover, the activation of YAP/TAZ is associated with chemoresistance in osteosarcoma cells to chemotherapeutic drugs such as methotrexate and doxorubicin. Activation of YAP/TAZ leads to decreased sensitivity to chemotherapeutic agents, thereby affecting treatment efficacy (*Wang et al., 2016b*).

The interaction between the Hippo signaling pathway and the tumor microenvironment promotes the malignant transformation of MSCs. Dysregulation of the Hippo pathway (*e.g.*, overexpression of TAZ) enhances the transformation of MSCs into cancer stem cells (CSCs), increasing tumorigenicity and chemoresistance (*Jabbari, Sarvestani & Moeinzadeh, 2015*). Telomere abnormalities lead to DNA damage and activate the p53 pathway (*Velletri et al., 2016*). If p53 function is lost, the combination of Hippo pathway dysregulation and telomere instability synergistically promotes genomic instability, accelerating tumorigenesis (*Zhou et al., 2019*).

### Hippo pathway in bone regeneration therapies

In recent years, in-depth investigations into the role of the Hippo signaling pathway in bone development and regeneration have revealed its potential therapeutic value for bone regenerative medicine (*Zhong, Jiao & Yu, 2024*). The Hippo pathway exerts critical regulatory effects on bone formation and repair by modulating cellular processes including proliferation, differentiation, and apoptosis (*Li et al., 2024b*). Although drugs directly targeting YAP/TAZ for osteoporosis treatment remain scarce, preclinical studies demonstrate that activating YAP/TAZ may confer partial therapeutic benefits. Estrogen deficiency represents a primary etiology of postmenopausal osteoporosis. Estrogen enhances osteoblast proliferation and differentiation while promoting bone formation through YAP/TAZ pathway-mediated regulation of Mechanical load in bone cells (*Shi & Morgan, 2024*). MSC-derived extracellular vesicles enriched with miR-23a-3p activate YAP/TAZ signaling *via* PTEN inhibition, thereby stimulating chondrocyte proliferation and bone tissue regeneration (*Piñeiro Ramil et al., 2025*). Verteporfin, a clinically approved drug, upregulates 14-3-3$\sigma$ protein expression to enhance cytoplasmic retention of YAP, thereby suppressing YAP/TAZ-TEAD complex formation and reducing transcriptional activity. This mechanism suggests its potential utility in modulating bone regeneration through context-dependent YAP/TAZ inhibition (*Wang et al., 2016a*). Multikinase inhibitors (*e.g.*, dasatinib and pazopanib), which target upstream YAP/TAZ activators such as SRC and FAK, have been shown to attenuate YAP/TAZ nuclear localization and activity. These agents may be repurposed to control YAP/TAZ-driven processes during bone regeneration (*Oku et al., 2015*). Statins, commonly prescribed for hypercholesterolemia,

interfere with YAP/TAZ signaling by modulating Rho GTPase activity. Notably, simvastatin has been shown to promote osteoblast differentiation while suppressing osteoclast activity, positioning it as a promising candidate for bone regenerative therapy (*Kamel et al., 2017*). Therapeutic strategies targeting YAP/TAZ hold considerable potential for treating bone disorders and enhancing osseous repair. Further research is warranted to fully elucidate the complex roles of YAP/TAZ in bone biology and to develop clinically translatable targeted therapies.

## CONCLUSIONS

In this review, the specific mechanism of the Hippo signaling pathway and its role as a key signaling pathway in bone-related diseases are explained. Abnormal regulation of the Hippo signaling pathway can cause metabolic imbalances in bone-related cells, such as osteoblasts and osteoclasts, and eventually induce the occurrence and development of bone metabolic diseases. YAP and TAZ, key downstream signaling molecules in the Hippo signaling pathway, are widely involved in bone diseases, including OA, spinal degenerative diseases, and osteosarcoma. As the key effector molecules in the Hippo signaling pathway, MST1/2, LATS1/2, YAP and TAZ can be used as regulatory sites of drugs. However, the specific upstream and downstream signaling molecules of these effector molecules are not well understood, and the exact regulatory mechanism has not been determined. How to further explore LATS1/2, YAP and other signaling molecules and kinases in the Hippo signaling pathway is the focus of future research. Further study of their specific interaction mechanisms will help us gain a deeper understanding of Hippo signaling pathway-mediated bone-related diseases and provide new treatment ideas. This finding has profound implications.

### Funding
This study was supported by the National Natural Science Foundation of China (32301416, 82372436, 82172468), the Yangzhou Lv Yang Jin Feng Program (YZLVJFJH2022YXBS154), and China Postdoctoral Science Foundation (2023T160553). The funders had no role in study design, data collection and analysis, decision to publish, or preparation of the manuscript.

### Grant Disclosures
The following grant information was disclosed by the authors:
National Natural Science Foundation of China: 32301416, 82372436, 82172468.
Yangzhou Lv Yang Jin Feng Program: YZLVJFJH2022YXBS154.
China Postdoctoral Science Foundation: 2023T160553.

### Competing Interests
The authors declare there are no competing interests.

## Author Contributions

- Chenwei Shao conceived and designed the experiments, performed the experiments, analyzed the data, prepared figures and/or tables, authored or reviewed drafts of the article, and approved the final draft.
- Hao Chen performed the experiments, authored or reviewed drafts of the article, and approved the final draft.
- Tingting Liu conceived and designed the experiments, analyzed the data, prepared figures and/or tables, and approved the final draft.
- Chun Pan conceived and designed the experiments, performed the experiments, prepared figures and/or tables, and approved the final draft.

## Data Availability

This is a literature review.

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
