# Peer review of "The Hippo pathway in bone and cartilage: implications for development and disease"

_PeerJ, doi:10.7717/peerj.19334_

## Round 0.1 · original submission · Major Revisions

When responding to all comments, please be sure to highlight the need for this paper and detail what it adds to the literature.

Reviewer 1 ·

Basic reporting

Thank you for the opportunity to review your article. The Hippo signaling pathway has been extensively studied in bone and cartilage, and its role in promoting bone regeneration is indeed significant. Currently, there is a wealth of research available on this topic. Regarding methodology, the researchers have conducted a relatively comprehensive search. However, while synthesizing the content, the article discusses bones and includes cartilage, which is not fully aligned with the title's focus on bones. Consider modifying the title to reflect the broader scope of the review.
Additionally, a review on the Hippo signaling pathway in bone homeostasis under the regulation of mechanics and aging has already been published. The authors could explore or summarize the pathway from other perspectives to distinguish this article. For instance, incorporating a discussion on the clinical translational applications of the Hippo pathway, such as its potential in bone regeneration therapies, would significantly enhance the impact of this review. Including more translational content would provide a unique highlight and differentiate this work from other reviews. Please cite: The Hippo signaling pathway in bone homeostasis: Under the regulation of mechanics and aging; Line 30 ,There is an additional Hippo here。
Lastly, there are multiple references to "Hippo" in the abstract, which should be streamlined for clarity and conciseness. The article is well-organized, but adding distinctive elements or highlights would make it more impactful and appealing. Thank you again for your contribution to this important area of research.

Experimental design

none

Validity of the findings

none

Additional comments

none

Reviewer 2 ·

Basic reporting

The paper entitled “The Hippo pathway in bone” is potentially interesting. The methods used were appropriate and the conclusions justified.
There are some possible issues 

 The paper discussed the role of Hippo pathway in bone. The signaling of hippo in osteoclasts is not fully shown.  Many studies have found the role of Hippo signaling pathway in regulating osteoclast formation (for example only, PMID: 29219182). It would be informative to discuss the role of The signaling of hippo in osteoclasts with a figure.


The mechanism of Hippo pathway in bone pathology is not clearly shown, for instance, the role of hippo in bone cancers such as osteosarcoma (for example only, PMID: 32697358) with Hippo signalling pathway for osteosarcoma therapies.

In Figure 2, some fonts are too small to see clearly.

It might be better to include a table listing all key signalling molecules and their relevant expression or presence in various bone cells, such as osteoclasts, osteoblast, chondrocytes?

Experimental design

see above

Validity of the findings

see above

Additional comments

see above

---

## Round 0.2 · accepted · Accept

The authors have addressed all concerns raised by the reviewers.

Reviewer 1 ·

Basic reporting

The author solve all the comment i have raised.

Experimental design

none

Validity of the findings

none

Additional comments

none

Reviewer 2 ·

Basic reporting

this a revised paper with questions addressed

Experimental design

this a revised paper with questions addressed

Validity of the findings

this a revised paper with questions addressed

Additional comments

this a revised paper with questions addressed